# INDIRECT ATTENTION: IA-DETR FOR ONE SHOT OBJECT DETECTION

## ABSTRACT

One-shot object detection presents a significant challenge, requiring the identification of objects within a target image using only a single sample image of the object class as query image. Attention-based methodologies have garnered considerable attention in the field of object detection. Specifically, the cross-attention module, as seen in DETR, plays a pivotal role in exploiting the relationships between object queries and image features. However, in the context of DETR networks for one-shot object detection, the intricate interplay among target image features, query image features, and object queries must be carefully considered. In this study, we propose a novel module termed "indirect attention". We illustrate that relationships among target image features, query image features, and object queries can be effectively captured in a more concise manner compared to cross-attention. Furthermore, we introduce a pre-training pipeline tailored specifically for one-shot object detection, addressing three primary objectives: identifying objects of interest, class differentiation, and object detection based on a given query image. Our experimental findings demonstrate that the proposed IA-DETR (Indirect-Attention DETR) significantly outperforms state-of-the-art one-shot object detection methods on both the Pascal VOC and COCO benchmarks.

## 1 INTRODUCTION

The field of object detection has seen remarkable advancements with the rise of deep learning technologies. However, the conventional approach of training models on a fixed set of classes presents significant limitations. Annotating all potential objects across diverse real-world environments is impractical, as existing systems are typically trained on a limited subset of objects. Scaling up this process is challenging. Few-Shot Object Detection (FSOD) addresses this challenge by detecting novel classes not seen during training, potentially overcoming many of the aforementioned limitations.

One-shot object detection (OSOD), a subset of FSOD, poses an even more demanding challenge, requiring the detection of objects within a target image using only a single sample image of the object class. This task is particularly challenging due to the necessity for models to generalize from extremely limited data. Attention mechanisms (Vaswani et al., 2017), especially self-attention and cross-attention, have become integral in capturing relationships within and between different data modalities. These mechanisms have been widely used across various domains, including multimodal learning (Bakkali et al., 2023) and one-shot and few-shot object detection (Lin et al., 2023). Recent advancements in attention-based methodologies, particularly the DETR (DEtection TRansformer) (Carion et al., 2020), have shown promise in object detection by leveraging the cross-attention mechanism to exploit relationships between object queries and image features. However, in the few-shot scenario, this correlation problem becomes more complex in DETR-based models due to the introduction of a third element, the object query. Recent DETR-based few-shot object detection methods, such as FS-DETR (Bulat et al., 2023) and Meta-DETR (Zhang et al., 2022), address this problem by incorporating an additional block of cross-attention, aligning target image features with query image features first and then passing the aligned features to the decoder for a second cross-attention with the object queries. The first feature alignment process between target image features and query image features, which initially seems essential for highlighting relevant areas in the target image based on the query, allows the detection head to focus on these areas during object detection. However, while this feature alignment strategy may seem necessary, it introduces

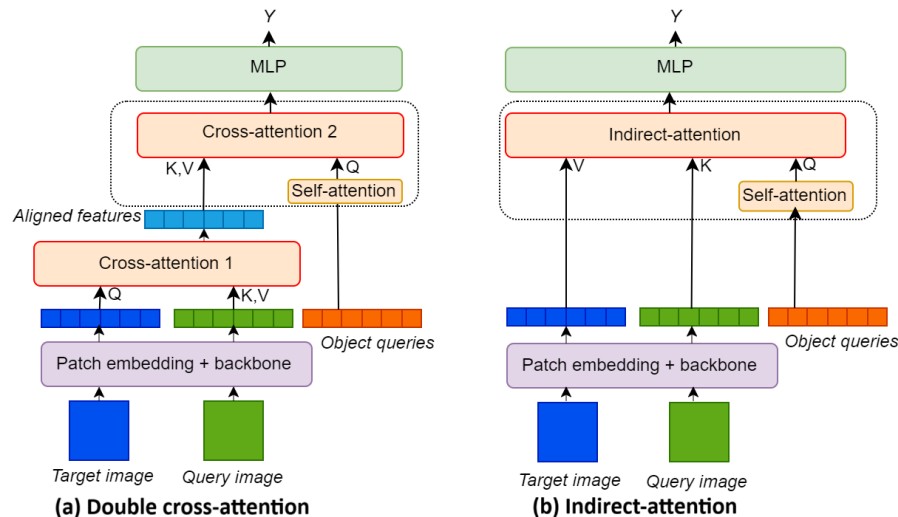

Figure 1: Illustration of the position of double cross-attention, feature alignment, and direct attention in the overall model.

significant computational overhead. The additional cross-attention block requires pairwise interactions between features from both images, leading to a quadratic increase in computational cost as the number of features grows. This burden becomes especially pronounced when dealing with high-resolution images, where the computational expense can severely limit the scalability and efficiency of the model.

To overcome this limitation, our work proposes a novel solution that challenges the need for explicit feature alignment. We introduce a new mechanism, which we term "indirect attention", that leverages the inherent strengths of the attention mechanism in transformers. Unlike traditional attention mechanisms, our indirect-attention uses inconsistent sequences for the key and value inputs, allowing the model to establish flexible interactions between the object queries, target image features, and query image features. By decoupling the key and value sequences, object queries can effectively extract information from the value vectors while relying on the key vector and a box relative position bias (Lin et al., 2023) which has been show to be important for performance in object detection in DETR, for guidance, all without direct feature alignment. This preserves the integrity of the features throughout the network and reduces the computational complexity. An illustration of the difference between double cross-attention and indirect-attention can be seen in figure 1.

A key innovation in our indirect-attention approach, which further departs from traditional attention mechanisms is utilizing inconsistent sequences for the key and value inputs. Typically, in standard attention mechanisms, the key and value sequences are aligned, ensuring that each query interacts with corresponding features in a consistent manner. However, in our method, we decouple this assumption and allow the key and value sequences to be distinct.

We also present a pre-training pipeline specifically designed for one-shot object detection, focusing on three primary objectives: identifying objects of interest, differentiating between classes, and accurately detecting objects based on the provided query image.

Our experimental results demonstrate that IA-DETR significantly outperforms existing state-of-the-art methods on prominent benchmarks such as Pascal VOC and COCO.

Our key contributions are summarized as follows:

- To our knowledge, in the field of object detection, we are the first to extend the transformer attention mechanism to three different elements, surpassing the traditional cross-attention's limitation of two elements.

- We apply our indirect attention mechanism to one-shot object detection, avoiding direct attention between the target image and query instance, thereby maintaining the integrity of both feature sets.
- IA-DETR outperforms the state-of-the-art in one-shot object detection on both the Pascal VOC and COCO datasets.

## 2 RELATED WORKS

### 2.1 ATTENTION MECHANISM

Attention has garnered significant interest since its introduction in (Vaswani et al., 2017) and has found applications across diverse domains. It can be conceptualized as a mapping between a query set and key-value pair sets, where the query is dynamically modified in the following manner:

$$\text{Attn} = \text{softmax}(\frac{QK^T}{\sqrt{d}})V,$$

where $Q$ denotes the query sequence, $K$ and $V$ denote the key and value sequences, receptively. In the self-attention mechanism, all queries, keys, and values are projections derived from the same input sequence. Mathematically, given an input sequence $S = [s_1, ..., s_n]$ with $n \geq 1$ and each element is of dimension $d$, the self attention can be formulated as:

$$\text{Self-Attn}(S) = \text{softmax}(\frac{W_q S S^T W_k^T}{\sqrt{d}})W_v S, \tag{1}$$

where $W_q$, $W_k$, and $W_v$ are learnable linear projections.

Cross-attention is another variation which has been used in few-shot object detection (Han et al., 2022). Mathematically, it can be formulated as:

$$\text{Cross-Attn}(S, M) = \text{softmax}(\frac{W_q S M^T W_k^T}{\sqrt{d}})W_v M, \tag{2}$$

where the $W_q$, $W_k$, and $W_v$ are learable linear projections. $S$ and $M$ are the two different input sequences. Note that the sequence $S$ serves as the query in the attention mechanism, while the sequence $M$ functions as both the key and the value.

### 2.2 ONE-SHOT OBJECT DETECTION

One-shot object detection aims to detect objects given only a single sample without fine-tuning. The model is trained only on base classes and then directly applied to detecting novel classes. SiamMask (Michaelis et al., 2018) enhances Mask R-CNN (He et al., 2017) by adding a matching module to generate a similarity feature map between the target and query images. CoAE (Hsieh et al., 2019) employs the non-local scheme (Wang et al., 2018) and squeeze-excitation scheme (Hu et al., 2018b) to correlate the target and query images. FOC OSOD (Yang et al., 2021) improves classification by decoupling the classification branch from the regression branch in both the RPN and detection head.

AIT (Chen et al., 2021) develops an attention-based encoder-decoder architecture with transformers (Vaswani et al., 2017) to evaluate the relationship between target and query images. BHRL (Yang et al., 2022) enhances alignment between the target image and query image by incorporating hierarchical and multi-scale feature attention. Unlike the aforementioned methods, UP-DETR (Dai et al., 2021), built upon DETR (Carion et al., 2020), adds the query image feature to the object query, after which the object query undergoes cross-attention blocks in the decoder module.

### 2.3 DETR AND ITS VARIANTS

DETR and its variants represent the application of transformers in object detection. DETR-based detectors primarily consist of a backbone, typically either a ResNet (He et al., 2016) or a Swin Transformer (Liu et al., 2021), followed by an encoder and a decoder. The encoder can be considered an extension of the backbone. However, recent work such as Plain-DETR (Lin et al., 2023) has

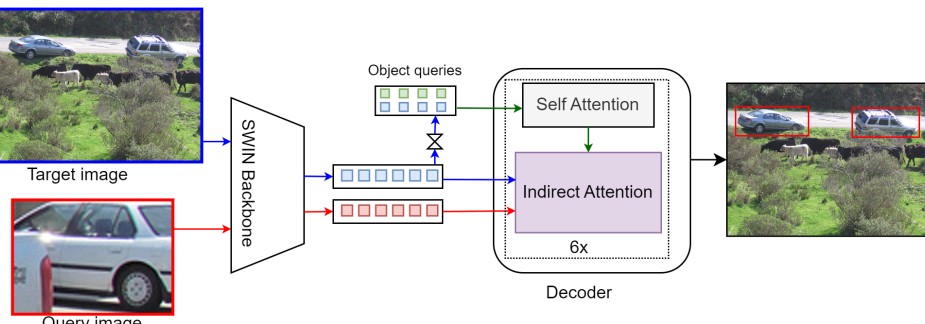

Figure 2: IA-DETR architecture.

demonstrated that the encoder is not always necessary. The decoder processes the output from the backbone and encoder, along with a set of object queries. These object queries pass through multiple decoder layers, undergoing self-attention and cross-attention with the backbone/encoder output to aggregate the necessary features for bounding box regression and classification. The self-attention mechanism in the decoder arranges the focus of the object queries, preventing them from concentrating on a single location, while cross-attention enables interaction with the original image features. In few-shot scenarios (Bulat et al., 2023), the architecture includes two cross-attention modules: one for aligning the query image features with the target image features, and another for exploiting the relationship between the object queries and the image features (both target and query images).

## 3 METHODOLOGY

### 3.1 PROBLEM DEFINITION OF ONE-SHOT OBJECT DETECTION

Given a training set consisting of seen classes $C_b$ and a test set containing new classes $C_n$ with $C_b \cap C_n = \emptyset$, the task of one-shot object detection is to train a detector on $C_b$ so that it can generalize to the test set and $C_n$ without additional training or tuning. Specifically, with a sample instance, also known as the query image $\mathbf{Q} \in \mathbb{R}^{H \times W \times 3}$ showing one instance of an object of a certain class, the detector is expected to display the bounding box $\mathbf{B} \in \mathbb{R}^4$ of all instances of the same class as $Q$ in the target image $\mathbf{I} \in \mathbb{R}^{H \times W \times 3}$, assuming the target image contains at least one instance of the same class as the object in $\mathbf{Q}$. This problem can also be viewed as a visual prompt task (Chen et al., 2024), where given the visual prompt $\mathbf{Q}$, the detector is expected to locate similar instances in the target image.

### 3.2 PROPOSED ARCHITECTURE

The architecture of the proposed IA-DETR is shown in Figure 3.2. Similar to other DETR-based models, it consists of a backbone and a decoder. Following Plain-DETR (Lin et al., 2023), we use SWIN-based MIM pretrained (Xie et al., 2022) as backbone and remove the DETR encoder, as the vision transformer-based backbone serves the same purpose. Both the query image and the target image are processed by the shared backbone. In the decoder, instead of using cross-attention, we propose indirect attention, which directly exploits the relationship between three elements: object queries, query image features, and target image features. Additionally, to avoid high computational costs, we use single-scale features for both target and query images, following the approach of Plain-DETR.

We also follow the iterative refinement approach as in (Zhu et al., 2020), where each decoder layer refines the bounding box predictions based on the output of the previous layer, rather than predicting them from scratch. The object queries are generated from the target image features without considering the query image. The rationale is that once potential objects are detected by analyzing the target image alone, these objects can later be filtered and refined in the decoder based on the query

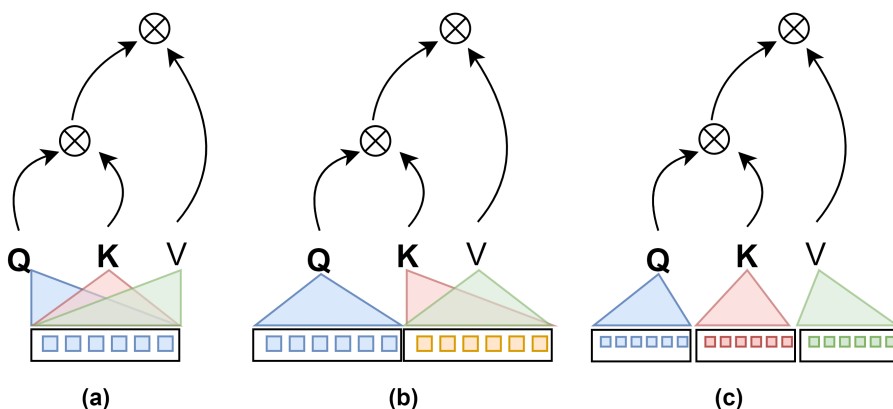

Figure 3: Illustration of difference between self-attention, cross-attention, and indirect-attention.

image. Instead of passing all proposals to the decoder, we select the top 300 object queries for the decoding process. Previous works (Jia et al., 2023; Hu et al., 2018a; Chen et al., 2022) have shown that the original one-to-one matching is less efficient in training positive samples, and that incorporating an auxiliary one-to-many set matching loss can improve efficacy. In one-shot object detection, where positive proposals are scarce, this technique can be particularly beneficial. Therefore, we have employed this hybrid matching technique during training.

### 3.3 INDIRECT ATTENTION

Cross-attention is typically used to exploit the relationships between two sequences, where one sequence acts as the query and the other sequence serves as both the key and the value. Although there has been limited exploration of this concept in computer vision, the idea of different entries for keys and values has emerged in the field of question answering, particularly in key-value memory networks (KVMNs) (Miller et al., 2016), where the key and value includes of different but related words, for example, in a sentence the subject is stored in key and the object is stored in value. Different from this, we propose indirect attention, which uses two different and potentially unrelated sources as the key, and value in the attention mechanism:

$$\text{Indirect-Attn}(S, M, L, B) = \text{softmax}\left(\frac{W_q S M^T W_k^T + B}{\sqrt{d}}\right) W_v L, \qquad (3)$$

where $S$, $M$, and $L$ are three different source sequences. $W_q$, $W_k$, and $W_v$ are learnable linear projections and $B$ is the relative positional bias.

The proposed indirect attention can be seen as a generalization of cross-attention, but with a crucial difference. While cross-attention functions as a matching and alignment module, where one sequence is aligned and modified based on information from another sequence, indirect attention modifies the sequence serving as the query based on its relationship with the value sequence while considering the key sequence, leaving the key and value sequences unchanged. An illustration of indirect-attention in comparison with self-attention and cross-attencion can be seen in figure 3

### 3.4 APPLICATION OF INDIRECT ATTENTION IN IA-DETR

The application of the proposed indirect attention in the context of one-shot object detection is straightforward. DETR models use object queries for localizing and classifying objects, necessitating consideration of the relationships between object queries, target image features, and query image features. Instead of using two cross-attention modules—one for aligning target image features with query image features and the other for exploiting the relationship between object queries and the image features (both target and query images)—the proposed indirect attention significantly simplifies this process.

Specifically, given a query image $\mathbf{Q} \in \mathbb{R}^{H \times W \times 3}$ and a target image $\mathbf{I} \in \mathbb{R}^{H \times W \times 3}$, both images are first encoded by a backbone encoder function $E$. This results in query image features $P = E(\mathbf{Q})$ and target image features $T = E(\mathbf{I})$. The object query $O^b$, where the superscript $b$ denotes the decoder block, then undergoes transformation within the decoder. More precisely, in the $b^{th}$ decoder block, the object queries are updated by a self-attention module as follows:

$$\hat{O}^b = O^b + \text{self-attn}(O^b).$$

The outputs $\hat{O}^b$ are further updated in the proposed indirect attention module with the help of both query image features $P$ and target image features $T$ as follows:

$$O^{b+1} = \text{FFN}(\hat{O}^b + \text{Indirect-Attn}(\hat{O}^b, P, T, B)). \tag{4}$$

Note that in the indirect attention, the query image features $P$ serves as the key vector and the target image features $T$ serves as the value vector.

In equation 4, we choose to use the Box-to-pixel relative postion bias (BoxRPB) as $B$ which is to compensate for the lack of multi-scale features (Lin et al., 2023). The use of BoxRPB guides attention to the areas of the bounding box for each object query (Lin et al., 2023). Therefore, it makes more sense to use the target image features as the value in indirect attention, rather than the aligned image features. We believe this is one of the key factors contributing to the strong performance of the proposed indirect attention for one-shot object detection.

For clarity purposes, the layer norm and drop-out are not shown in the equations

### 3.5 TRAINING STRATEGY

We design a training strategy specifically for one-shot object detection. The core idea is to train the network to effectively perform three tasks: identifying objects of interest, detecting objects based on a given query, and differentiating between classes. The first two tasks follow a coarse-to-fine detection approach, which aligns with the methodology of most object query-based detectors. The third task, class differentiation, is a common goal in classification problems but is often overlooked in one-shot object detection, especially during the pre-training stage.

In this paper, we adopt a commonly used two-stage training approach and propose incorporating contrastive loss (Xie et al., 2021) in the pre-training stage. We argue that this inclusion enhances the model's performance in one-shot object detection.

In the first stage (pretraining stage), the model is trained in a supervised manner, enhanced by a self-supervised approach. Specifically, images containing only seen classes are selected. For each image, similar to (Dai et al., 2021) a random patch is cropped, with its position used during training. This patch serves as the query image, which is zero-padded and input into the model along with the original image as the target image. The model is trained to simultaneously localize the query patch and detect all objects in the target image.

In the existing methods, pretraining is done on classification task which is different from the object detection problem. More precisely, a ground truth vector is constructed, containing objectness and bounding box information for all objects and the query patch. In this vector, the objectness of each object is set to 1, while the objectness of the query patch is set to 0. The detection result for each object query includes an objectness class (0 or 1) and a predicted bounding box. A one-to-one matching process between the detection results and the ground truth is performed, a common practice in DETR-based detectors (Sun et al., 2021). Given that the number of object queries typically exceeds the number of objects in an image, unmatched detections are considered background. Subsequently, classification loss and bounding box loss are calculated as per standard procedures (Lin et al., 2023).

We propose incorporating a contrastive loss (Xie et al., 2021) using ground truth class information. For each object query matched to ground truth objects, we embed the bounding box in the detection result and apply an additional contrastive loss. The goal is to bring embeddings of detections from the same class closer together and push embeddings from different classes further apart. Ground truth class information is used to construct positive and negative sets, and the contrastive loss is calculated accordingly. Background detections and query patch are excluded from this calculation. For each predicted bounding box feature embedding $\hat{p}_i$ corresponding positive and negative box

feature embeddings $P_+$ and $P_-$ are selected based on being in the same class as $\hat{p}_i$ and the contrastive loss is caluculated with $\tau$ as temperature hyper-parameter that controls the difficulty of the task of contrastive learning (Wang & Liu, 2021) as:

$$\mathcal{L}_{\text{con}} = -\sum_{\hat{p}_i} \sum_{P_+} \log \frac{\exp(\hat{p}_i \cdot P_+/\tau)}{\exp(\hat{p}_i \cdot P_+/\tau) + \sum_{P_-} \exp(\hat{p}_i \cdot P_-/\tau)} \tag{5}$$

The overall loss of the pre-training stage is:

$$\mathcal{L} = \lambda_1 \mathcal{L}_{\text{bbox}} + \lambda_2 \mathcal{L}_{\text{cls}} + \lambda_3 \mathcal{L}_{\text{con}}, \tag{6}$$

where $\mathcal{L}_{\text{bbox}}$, $\mathcal{L}_{\text{cls}}$, and $\mathcal{L}_{\text{con}}$ are the bounding box, classification, and contrastive losses with their relevant factors $\lambda$. The bounding box and classifcation losses are computed same as in (Lin et al., 2023). It is also important to note that throughout the first stage, the backbone remains frozen.

During the second stage, the model is trained for standard one-shot object detection, aiming to detect instances of the query image within the target image. The loss function in this stage comprises bounding box loss and classification loss, calculated in the same manner as described in (Lin et al., 2023).

In the second stage, we unfreeze the backbone initially but freeze it again after the second epoch to prevent overfitting on the seen classes.

## 4 EXPERIMENTS

### 4.1 DATASETS AND METRICS

For the purpose of fair comparison, we follow the same OSOD dataset setting as the previous works (Chen et al., 2021; Yang et al., 2022). In PASCAL VOC dataset (Everingham et al., 2010), we devide the 20 classes to two sets of 16 seen classes and 4 unseen classes. For the MS COCO dataset (Lin et al., 2014), 4 splits are created in a way that the 80 classes are equally divided into 4 parts (20 classes per part) and in each split three parts serves as unseen classes while the other one part serves as seen classes. Following (Chen et al., 2021; Yang et al., 2022) we report AP-50 for both PASCAL VOC and COCO datasets.

### 4.2 IMPLEMENTATION DETAIL

For the first stage training, we train the model for 30 epochs with batch size of 24 on 4 GPUs using the SGD optimizer. In this stage, we keep the backbone frozen and only train the decoder part.

In the second stage, we train the model for 14 epochs with batch size of 16 on 8 GPUs using the SGD optimizer. During both stages, the model is trained only on seen classes.

### 4.3 TARGET AND QUERY PAIRS

In the first stage, the query is generated same as UP-DETR (Dai et al., 2021) by cropping a random part of the target image and use it as the query image.

In the second stage, we follow previous OSOD works (Chen et al., 2021; Yang et al., 2022; Hsieh et al., 2019) to generate the target-query image pairs. During training, for a given target image containing an object from a seen class, we randomly select a patch of the same class from a different image. During testing, for each class in the target image, query patches of the same class are shuffled using a random seed set to the image ID of the target image. The first five patches are then selected, and the average metric score is reported.

### 4.4 QUANTITATIVE RESULTS

In Table 1, we compare the performance of IA-DETR with state-of-the-art methods on the Pascal VOC dataset for both seen and unseen classes. The results clearly show that IA-DETR significantly outperforms existing methods in both categories.

Table 1: Comparison results on Pascal VOC dataset. Results based on $AP_{0.5}$.

| Method | plant | sofa | tv | car | bottle | boat | chair | person | bus | train | horse | bike | dog | bird | mbike | table | Avg. | cow | sheep | cat | aero | Avg. |
|---|---|---|---|---|---|---|---|---|---|---|---|---|---|---|---|---|---|---|---|---|---|---|
| | | | | | | | | **Seen classes** | | | | | | | | | | **Unseen classes** | | | | |
| SiamFC (Bertinetto et al., 2016) | 3.2 | 22.8 | 5.0 | 16.7 | 0.5 | 8.1 | 1.2 | 4.2 | 22.2 | 22.6 | 35.4 | 14.2 | 25.8 | 11.7 | 19.7 | 27.8 | 15.1 | 6.8 | 2.28 | 31.6 | 12.4 | 13.3 |
| SiamRPN (Li et al., 2018) | 1.9 | 15.7 | 4.5 | 12.8 | 1.0 | 1.1 | 6.1 | 8.7 | 7.9 | 6.9 | 17.4 | 17.8 | 20.5 | 7.2 | 18.5 | 5.1 | 9.6 | 15.9 | 15.7 | 21.7 | 3.5 | 14.2 |
| OSCD (Fu et al., 2021) | 28.4 | 41.5 | 65.0 | 66.4 | 37.1 | 49.8 | 16.2 | 31.7 | 69.7 | 73.1 | 75.6 | 71.6 | 61.4 | 52.3 | 63.4 | 39.8 | 52.7 | 75.3 | 60.0 | 47.9 | 25.3 | 52.1 |
| CoAE (Hsieh et al., 2019) | 24.9 | 50.1 | 58.8 | 64.3 | 32.9 | 48.9 | 14.2 | 53.2 | 71.5 | 74.7 | 74.0 | 66.3 | 75.7 | 61.5 | 68.5 | 42.7 | 55.1 | 78.0 | 61.9 | 72.0 | 43.5 | 63.8 |
| AIT(Chen et al., 2021) | 46.4 | 60.5 | 68.0 | 73.6 | 49.0 | 65.1 | 26.6 | 68.2 | 82.6 | 85.4 | 82.9 | 77.1 | 82.7 | 71.8 | 75.1 | 60.0 | 67.2 | 85.5 | 72.8 | 80.4 | 50.2 | 72.2 |
| UP-DETR(Dai et al., 2021) | 46.7 | 61.2 | 75.7 | 81.5 | 54.8 | 57.0 | 44.5 | 80.7 | 74.5 | 86.8 | 79.1 | 80.3 | 80.6 | 72.0 | 70.9 | 57.8 | 69.0 | 80.9 | 71.0 | 80.4 | 59.9 | 73.1 |
| BHRL(Yang et al., 2022) | 57.5 | 49.4 | 76.8 | 80.4 | 61.2 | 58.4 | 48.1 | 83.3 | 74.3 | 87.3 | 80.1 | 81.0 | 87.2 | 73.0 | 78.8 | 38.8 | 69.7 | 81.0 | 67.9 | 86.9 | 59.3 | 73.8 |
| IA-DETR | 39.3 | 69.4 | 78.3 | 82.7 | 52 | 73.7 | 49.8 | 52.6 | 86.6 | 86.3 | 92.4 | 86.7 | 90.4 | 88.2 | 79.9 | 69.5 | **73.6** | 90.5 | 81.2 | 85.2 | 67.4 | **81** |

Table 2: Comparison results on MS COCO dataset. Results are based on $AP_{0.5}$.

| Method | Seen classes | | | | | Unseen classes | | | | |
|---|---|---|---|---|---|---|---|---|---|---|
| | split-1 | split-2 | split-3 | split-4 | Average | split-1 | split-2 | split-3 | split-4 | Average |
| SiamMask (Michaelis et al., 2018) | 38.9 | 37.1 | 37.8 | 36.6 | 37.6 | 15.3 | 17.6 | 17.4 | 17.0 | 16.8 |
| CoAE (Hsieh et al., 2019) | 42.2 | 40.2 | 39.9 | 41.3 | 40.9 | 23.4 | 23.6 | 20.5 | 20.4 | 22.0 |
| AIT (Chen et al., 2021) | 50.1 | 47.2 | 45.8 | 46.9 | 47.5 | 26.0 | 26.4 | 22.3 | 22.6 | 24.3 |
| BHRL (Yang et al., 2022) | 56.0 | 52.1 | 52.6 | 53.4 | 53.5 | 26.1 | 29.0 | 22.7 | 24.5 | 25.6 |
| IA-DETR | 53.2 | 55.6 | 56.2 | 58.1 | **55.8** | 27.3 | 27.0 | 28.7 | 26.4 | **27.3** |

To further validate the superiority of IA-DETR, we evaluate our model against other methods on the challenging COCO dataset across all four splits. The results, presented in Table 2, demonstrate that IA-DETR consistently outperforms all existing methods by an average of 2% on both seen and unseen classes.

## 5 ABLATION STUDIES AND ANALYSIS

In this section, we conduct extensive experiments to study the behavior of different components of IA-DETR and indirect-attention. All experiments are performed on the Pascal VOC dataset.

First, a natural question may arise: what if we explore different variations of the roles assigned to object queries, target image features, and query image features as the query, key, and value in the indirect-attention mechanism? While the role of object queries as the query is inherently fixed, permutations of the target image and query image features as the key and value are worth investigating. However, empirical results demonstrate that setting the query image features as the value and the target image features as the key yields zero performance, even with extended training durations. This outcome aligns with intuitive reasoning: the target image features are best suited as the value, as they ultimately serve as the source from which object bounding boxes and class predictions are extracted.

We compare the proposed indirect attention method with two configurations of dual cross-attention layers, as this is a common approach. In the first configuration, the goal is to first match the target image $T$ to the query image $Q$ using cross-attention. Then, a second cross-attention is performed between the object queries $O$ and the output of the first cross-attention, treating it as merged features as follows:

$$F_m = \text{Cross-attn}(T, Q),$$
$$O = \text{cross-attn}(O, F_m).$$

In the second configuration, we maintain two cross-attention blocks, where the object queries serve as the query in the second block, the output of the first cross-attention block acts as the key, and the original target image features serve as the value. We conduct ablations on both cross-attention configurations alongside indirect-attention on the Pascal VOC dataset, without the 1st stage of training and early freezing of the backbone.

As depicted in Table 4, indirect-attention outperforms both cross-attention configurations while utilizing only one attention block. We attribute this to the fact that in indirect-attention, the object query has access to both target and query image features.

In Table 5, we investigate the significance of BoxRPB on indirect-attention. Given the absence of direct spatial relationship between key and value, it's crucial to assess if the query can discern

semantic associations between them. We observe that without BoxRPB, the performance of the model drops significantly and even with continued training over several more epochs it does not match the performance of its counterpart.

In order to further study the effect of BoxRPB by removing it from the model with double cross-attention and BoxRPB only. The BoxRPB only mode does not involve any kind of interaction between target features and query features. As the result can be observed in BoxRPB the double cross-attention and BoxRPB only mode do not get a big drop in performance. However still the combination of indirect-attention with BoxRPB results to the best performance with fewer number of parameters.

Table 3: Experiment results on BoxRPB. Results are based on $AP_{0.5}$.

| double cross-attention | BoxRPB | indirect-attention | Seen classes | Unseen classes | #param. |
|---|---|---|---|---|---|
| ✓ | ✓ | ✗ | 83.61 | 63.34 | 69M |
| ✓ | ✗ | ✗ | 77.9 | 62.31 | 69M |
| ✗ | ✓ | ✗ | 81.93 | 61.31 | 60M |
| ✗ | ✓ | ✓ | 82.94 | 65.13 | 61M |

In Table 6, we delve into the importance of our two-stage training strategy. Notably, the contrastive loss in first stage of training and early freezing of the backbone yield significant performance improvements. Specifically, early freezing of the backbone aids in generalization across both seen and unseen classes and creating a balance between the seen and unseen classes. The performance on the unseen classes increase but at the cost of decrease in the seen classes. This can be seen in a way that the less the backbone overfits on the seen classes the better it can generalize on unseen classes. On the other hand, the MIM pretraining backbone though contributes in performance enhancement but it is not very significant. However, such a pretraining strategy for the backbone is useful as it does not rely on labeled data and alleviates the problem of limited labeled data availability. Additionally, these findings underscore the efficacy and generalizability of indirect-attention, highlighting its independence from specific features learned by the backbone.

## 5.1 VISUALIZATION

To comprehend the behaviors of indirect-attention, we conducted extensive visualization of the attention maps. Through our analysis, we made the following observations:

- Certain attention heads prioritize the content of the query image features, while others concentrate on specific locations within the target image features.

- The indirect attention mechanism selects object queries based on the conditioning provided by the query image features.

To investigate how queries are ranked, purely for visualization purposes, we compute the output of the dot product between the query and key. Although the model applies softmax along the key dimension, for query ranking understanding, we reverse the softmax operation by performing softmax along the query dimension. It's noteworthy that not all attention heads focus on the key (query image), only specific heads do. We extract values along these specific heads, averaging them. Then, we average again along the key dimension, resulting in a vector with the same length as the number

Table 4: Comparison of indirect-attention with two configurations using double cross-attention blocks. Results are based on $AP_{0.5}$.

| Method | Seen | Unseen |
|---|---|---|
| double cross-atten. 1 | 83.61 | 63.34 |
| double cross-atten. 2 | 82.34 | 63.21 |
| indirect-attention | 82.94 | 65.13 |

Table 5: Effect of removing BoxRPB on indirect-attention. Results are based on $AP_{0.5}$.

| Method | Epochs | Seen | Unseen |
|---|---|---|---|
| w/o BoxRPB | 14 | 29.21 | 33.8 |
| w/o BoxRPB | 80 | 70 | 58.6 |
| with BoxRPB | 14 | 82.94 | 65.13 |

Table 6: Impact of MIM pretraining of backbone, contrastive loss in 1st stage, and early backbone freezing in 2nd stage. The backbone freezing in second stage is done after the second epoch. Results are based on $AP_{0.5}$

| MIM pre-trained backbone | contrastive loss | early backbone freezing | Seen | Unseen |
|---|---|---|---|---|
| ✓ | ✗ | ✗ | 82.63 | 64.81 |
| ✓ | ✓ | ✗ | 82.94 | 65.13 |
| ✗ | ✓ | ✓ | 74.35 | 79.5 |
| ✓ | ✓ | ✓ | 73.6 | 81 |

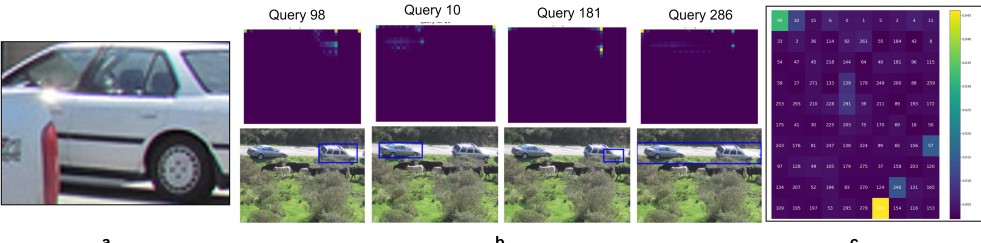

Figure 4: (a): Query image. (b): attention map and detected objects of 4 object queries. (c): attention score based on object queries. Number in each cell shows the query id.

of object queries. This vector is ordered based on the model's confidence score in the final prediction. To enhance visualization, we reshape the ordered vector into a two-dimensional matrix. As illustrated in figure 4, object queries related to the query image object receive higher attention than others.

### 5.2 POTENTIAL LIMITATIONS

While the technique presented offers an effective approach for one-shot object detection and provides an efficient alignment solution, there are notable limitations we wish to address:

- The indirect attention method severs the direct alignment between the key and value, relying extensively on relative positional bias to steer the attention matrix to the appropriate position within the value sequence.

- OSOD methods presuppose the presence of at least one instance of the query image in the target image, which may not always hold true in real-world scenarios.

## 6 CONCLUSION

One-shot object detection holds significant importance in real-world scenarios where obtaining sufficient annotated data for training is challenging. In this paper, we introduce indirect attention as a viable alternative to two cross-attention blocks, demonstrating the capability of the transformer attention mechanism to accommodate three different sequences as input. Building upon indirect attention, we propose IA-DETR, which meticulously considers the complex relationship between object queries, target images, and query images within a single indirect-attention module. Our approach achieves state-of-the-art results on both the Pascal VOC and MS COCO datasets.

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
