# OpenReview forum: "INDIRECT ATTENTION: IA-DETR FOR ONE SHOT OBJECT DETECTION"
_ICLR.cc/2025/Conference — ICLR 2025 Conference Withdrawn Submission_

### Official Review · Reviewer_As9n · 2024-10-31

**Soundness:** 2
**Presentation:** 2
**Contribution:** 2
**Rating:** 5
**Confidence:** 4

**Summary:**

The paper addresses the limitations of the double cross-attention strategy found in current one-shot object detection (OSOD) methods and presents the indirect-attention (IA) module as a practical alternative. In addition to the IA, the proposed IA-DETR model enhances the OSOD task by incorporating Box-to-pixel relative position bias (BoxRPB) and a contrastive pre-training pipeline specifically designed for the one-shot object detection head. Experimental results demonstrate the effectiveness of this model on the Pascal VOC and COCO datasets.

**Strengths:**

+ The concept of indirect attention is straightforward and can be easily understood and implemented.

**Weaknesses:**

- The experiments conducted may have issues regarding fairness. When evaluating the effectiveness of various OSOD methods, the choice of backbone architecture is significant. The proposed IA-DETR utilizes SWIN-based MIM pre-trained weights as its backbone, which differs from the more commonly used ResNet50 and reduced-ImageNet pre-trained weights in existing OSOD methods. It would be beneficial to first validate the proposed model architecture with the same backbone before progressing to a stronger one. Additionally, it's important to note that in the OSOD task, the dataset used for obtaining the pre-trained weights should exclude any classes that are present in the Pascal VOC and COCO datasets.

- The authors assert that the double cross-attention block results in a quadratic increase in computational cost as the number of features increases. However, their experiments do not provide sufficient support or clarification on how this increased computational burden impacts an OSOD model.

- In the proposed IA-DETR, it is interesting to consider alternative combinations of object queries, query image features, and target image features for their roles as query, key, and value. Including comparisons of these variations in the ablation study would enhance the comprehensiveness of the research.

- The findings from the ablation study indicate that the performance improvements attributed to the proposed indirect attention mechanism and the contrastive pre-training pipeline are quite modest. It appears that the overall effectiveness of the model is more significantly influenced by the backbone and the BoxRPB component. Consequently, the technical contributions of these enhancements are somewhat constrained.

**Questions:**

The primary concern of this manuscript is on the experiments conducted. It is important to assess the appropriateness of utilizing the SWIN-based MIM pretrained backbone. Additionally, the ablation study does not adequately demonstrate the contributions of the indirect attention mechanism and the contrastive pre-training pipeline as claimed by the authors. To enhance the manuscript, a more detailed analysis should be included to clarify the effects of the proposed indirect attention and the contrastive pre-training approach.

---

> ### Author Response · Authors · 2024-11-21
>
> Thank you for your critical review. We address your concerns with additional empirical evidence that demonstrates the robustness of indirect-attention:
> 1. Thanks for mentioning the ResNet50 backbone. Here is the result with ResNet50 backbone on pascal VOC, pretrained on reduced imagenet:
> | Method | Seen Classes | Unseen Classes |
> |---|---|---|
> | BHRL | 69.7 | 73.8 |
> | IA-DETR with ResNet50 | **77.8** | **79.56** |
>
> 2. The quadratic scaling of computational complexity with image size is inherent to transformers attention mechanisms. However, our indirect attention is more efficient than double cross-attention by design - it requires half the attention blocks while achieving superior performance. Following is further results on computational complexity comparing both double cross attention setting with indirect-attention.
> | Method | Image Size | FLOPs (G) | Memory (GB) |
> |---|---|---|---|
> | Double Cross-Attention | 512x640 | 186.3 | 9.7 |
> | Double Cross-Attention | 1024x1024 | 536.3 | 26.7 |
> | Indirect Attention | 512x640 | **173.7** | **9.4** |
> | Indirect Attention | 1024x1024 | **478.2** | **23.3** |
>
> 3. We have tried different permutations of object queries, target image features, and query image features as key, query, and value. However, the object queries need to be the query but the target and query image features can be permuted. We have tried a different variation by setting target image features as key, and query image features as value but the model eventually fails to learn anything even during extended training time. Intuitively also it makes sense for the target image feature to be set as value and not as key because finally since they're the source for final box and class predictions.
>
> 4. The interplay between BoxRPB and indirect attention is an important consideration that our ablation studies help clarify. As shown in row 3 of Table 3, while BoxRPB contributes to performance and is important for indirect-attention, indirect attention plays a crucial role - particularly for unseen classes where we observe significant performance downgrade in the case of removal of indirect-attention and relying only on BoxRPB. It is worthy to mention that BoxRPB is an enhanced position bias[1, 2] specific for object detection. Similarly, Table 6 shows performance improvement by adding the contrastive loss though not very substantial.
>
> Questions:
> We hope the explanations and further experiment results provided above can answer the concerns.
>
> [1]: Bao, Hangbo, et al. "Unilmv2: Pseudo-masked language models for unified language model pre-training." International conference on machine learning. PMLR, 2020.
>
> [2]: Liu, Ze, et al. "Swin transformer: Hierarchical vision transformer using shifted windows." Proceedings of the IEEE/CVF international conference on computer vision. 2021.

---

> ### Comment · Reviewer_As9n · 2024-11-27
> **Respond to the Rebuttal**
>
> Thanks to the author's effort in providing a rebuttal.
> 1. Upon examining the results, it is intriguing that ResNet50, when trained on the reduced Imagenet, can outperform the original IA-DETR using SWIN trained on the full Imagenet. It is better to clarify the reasoning behind this unexpected outcome, particularly how a model with a seemingly weaker backbone and less data achieves superior results on the seen classes. Additionally, how is the performance of IA-DETR with ResNet50 being assessed in the COCO dataset?
> 2. Thanks for the information.
> 3. It is recommended supporting these findings with quantitative experimental results and comparing them to the traditional double cross-attention method.
> 4. It is also noteworthy that Table 3 highlights that the role of BoxRPB is more significant than that of direct-attention.

---

> ### Author Response · Authors · 2024-11-27
>
> Dear reviewer,
> Thanks a lot for your response.
> 1. Thank you for your thoughtful feedback and for highlighting this intriguing observation. Comparing backbones of different architectures trained on distinct tasks is indeed challenging. Our explanations for this issue are as followings:\
> **Pretraining objectives of two backbones and dataset characteristics**: Firstly, ResNet50 is trained on classification task and the objects in PascalVOC are not dense compared to COCO dataset. Most of the images in PascalVOC have a single big object in the middle making the classification part of the problem heavier than the localization part which makes it well algined with ResNet-50's pretraining which is mainly classification. On the other hand, it is true that SWIN has been trained on full Imagenet but with masked image modelling task without seeing the class labels, which makes it different. In fact, one can say that SWIN has also been trained on reduced Imagenet considering that the class labels are an important part of the dataset.\
> **Seen vs. unseen classes generalization tradeoff**: Secondly, based on our observations, the higher performance on seen classes can be explained by the lower performance on unseen classes and generalizing on unseen classes demands sacrifice of performance on seen classes. In other words, the better the backbone overfits on seen classes the worse it will get on unseen classes. This can be supported by table-6 in our experiments where it can be seen that performance on unseen classes increases substantially when we apply early backbone freezing but at the cost of drop on seen classes. We have expanded slightly on this in our new revision.\
> To support our first point and as suggested by the reviewer we have done a limited test of the Resnet50 backbone **only on split-1** of COCO (60 class as seen and 20 class as unseen). Unlike PascalVOC in here we do not see very high performance on seen classes for Resnet-50:
> | Method | Seen Classes | Unseen Classes |
> |---|---|---|
> | IA-DETR with ResNet50 | 52.6 | 26.5 |
> | IA-DETR with SWIN | **53.2** | **27.3** |
>
> 3. Thanks for recommending this. In our updated submission we have reflected on this in the beginning of the experiments section. The quantitative result for a different combination (query image as value, target image as key) is simply 0 for both seen and unseen classes even if trained for more epochs. It is important to mention that the roles of object queries, query image and target image features are fixed in double cross-attention setting so we can’t make comparisons in this regard.
>
> 4. We do agree on this and we have mentioned this as a possible limitation in subsection 5.2 as the first point.
>
> We hope that the provided clarifications resolve the concerns.

---

### Official Review · Reviewer_h13P · 2024-11-01

**Soundness:** 3
**Presentation:** 3
**Contribution:** 3
**Rating:** 8
**Confidence:** 4

**Summary:**

The paper proposed a novel DETR structure named "IA-DETR" that aims to more effectively capture the relationship among the target image features, query image features, and object queries. The method is evaluated on the standard PASCAL and COCO datasets, and the experimental results indicate that the proposed method has surpassed the existing methods.

**Strengths:**

1. The proposed method is novel and may have a positive effect on other related research.
2. The experimental results indicate that the method has achieved SOTA on the standard benchmark.
3. The motivation is clear, and the paper is in good structure.

**Weaknesses:**

1. Although the motivation is clearly stated, the first two paragraphs are slightly tedious. The author could consider trimming this introduction and making the motivation more straightforward.
2. In line #077 ~ #082, the paper states that one motivation of the proposed method is to ease the computational overhead in an existing method caused by additional cross-attention. To support this assumption, it is necessary to include an ablation study regarding the computational expense. If I missed this, please point it out during the rebuttal.
3. According to recent works on DETR (e.g., SQR [Chen et al.]), it is not only the final layer of the decoder that produces the correct prediction results; the output of the middle layer of the decoder sometimes produces better results. Is the proposed indirect-attention applied to each layer of the decoder? Is it possible that only applying on several layers of the decoder would get better performance?
4. In the last two rows of Table 6, the MIM pre-trained backbone decreases the AP50 of seen categories but increases the unseen categories. While the paper claims that the MIM is not very significant, then there should be more discussion about why the MIM is still necessary here.


[Chen et al.] Enhanced Training of Query-Based Object Detection via Selective Query Recollection, CVPR, 2023

**Questions:**

Please refer to the weakness.

---

> ### Author Response · Authors · 2024-11-21
>
> Thanks a lot for your feedback and guidance.
> 1. Thanks for suggesting that, we will enhance the introduction part in the final version.
>
> 2. We provide the number of parameters in both (double cross-attention and indirect attention) settings in table 3. And the computation complexity comparison of both is as following:
> | Method | Image Size | FLOPs (G) | Memory (GB) |
> |---|---|---|---|
> | Double Cross-Attention | 512x640 | 186.3 | 9.7 |
> | Double Cross-Attention | 1024x1024 | 536.3 | 26.7 |
> | Indirect Attention | 512x640 | **173.7** | **9.4** |
> | Indirect Attention | 1024x1024 | **478.2** | **23.3** |
>
> 3. Yes the all the blocks in decoder part are based on indirect-attention.
> 4. Thanks for pointing this out, we have expanded on this considering the fact that the MIM pretraining is better because it does not rely on labelled images and allow for pre-training on large pile of images and alleviate the problem of limited labelled data availability that OSOD tries to tackle.

---

> > ### Comment · Reviewer_h13P · 2024-11-22
> > **Thanks for the clarification**
> >
> > The authors have addressed most of my concerns. Though I'm still hesitant about the Q3, but overall, I think the authors have done substantial revising to their manuscript. And I approve their claim that "we challenge and overturn the long-standing assumption that attention keys and values must be aligned".

---

> > > ### Author Response · Authors · 2024-11-24
> > >
> > > Appologies that we did not notice the second part of question #3 at first. In fact, due to the mentioned issue that the intermediate blocks in decoder may produce better result, the subsequent works on DETR follow iterative refinement[1] in which each decoder block predicts deltas on the total predicted box up to the block, and the box loss is calculated on each block of the decoder. However, the idea to use indirect-attention in some of the blocks seems interesting. Thus, in a new experiment we used indirect-attention in the three first blocks and switched to normal cross-attention (between object queries and target image feature) in the three last blocks (there are total of 6 blocks). However the performance drops on unseen classes. Followings are the result compared with when all decoder blocks use indirect-attention:
> > > | Method | Seen Classes | Unseen Classes |
> > > |---|---|---|
> > > | IA in first 3 blocks only | 81.72 | 61.15 |
> > > | IA in all blocks | **82.94** | **65.13** |
> > >
> > > [1]: Zhu, Xizhou, et al. "Deformable detr: Deformable transformers for end-to-end object detection." ICLR (2021).

---

> > > > ### Comment · Reviewer_h13P · 2024-11-25
> > > > **Thanks for the clarification**
> > > >
> > > > Thanks, that's an interesting observation, which also strengthen the effectiveness of IA, and I believe it addressed my concern.

---

### Official Review · Reviewer_xquQ · 2024-11-01

**Soundness:** 2
**Presentation:** 3
**Contribution:** 2
**Rating:** 5
**Confidence:** 4

**Summary:**

Overall: This paper introduces IA-DETR, a novel one-shot object detection model that uses indirect attention to efficiently capture relationships between target image features, query image features, and object queries. The proposed method significantly outperforms state-of-the-art techniques on the Pascal VOC and COCO benchmarks, demonstrating its effectiveness and efficiency.

**Strengths:**

- Good formulas and figures.
- The idea of extending the transformer attention mechanism to three distinct elements is simple.
- Comprehensive experiments yield robust and state-of-the-art results.
- The concept of IA-DETR is innovative for OSOD (One-Shot Object Detection).

**Weaknesses:**

-  Novelty is limited: the new technical thing proposed in this paper is "indirect attention" which differs from the previous attention by using two inputs for K and V.   However, this idea seems direct and too simple without other technical contributions.
- The experimental analysis of the indirect attention is not comprehensive, such a manner could be regarded as using the K, and V layers to fuse the features of the input K (query images features P) and V (target image features T), how about the comparison result of first using some other simple fusion methods e.g., MLP([P, T])  and then the typical cross-attention.
- The experiments in Tables 1 and 2 are not based on multiple runs, which will weaken the robustness of the proposed method.
- The paper does not explicitly state whether the indirect attention method is applied during the pre-training stage. Given that the main challenge in OSOD is the scarcity of positive samples, and the proposed method succeeds during fine-tuning, it should ideally also be effective in the pre-training stage, where there are more positive samples available. Therefore, if indirect attention is applied during pre-training, results for this stage should also be presented
-  Minor: Figure 3 is too large.
- The left and right quotation marks do not match on lines 18 and 83.

**Questions:**

please see weakness

---

> ### Author Response · Authors · 2024-11-21
>
> We thank the reviewer for the comments.
> 1. While our indirect attention mechanism may appear straightforward, this simplicity masks a fundamental contribution: we challenge and overturn the long-standing assumption that attention keys and values must be aligned. This implicit constraint has gone unquestioned in attention mechanisms since their inception. Breaking free from this assumption enables a new class of attention operations empirically validated on and particularly suited for OSOD. Although the precise theoretical underpinnings of this mechanism require further exploration, we attempted to gain insights through the visualizations presented in Section 5.1.
>
> 2. The interpretation of indirect attention as simple feature fusion between K, and V offers an interesting perspective, but we believe the mechanism operates quite differently. In standard attention, K and V represent different embeddings of the same sequence, operating as distinct projections rather than elements to be fused. Our indirect attention extends this principle: K and V maintain their independent roles through multiple attention blocks, just as in standard attention, with the key innovation being their origin from different sources. We conduct the experiment with an alignment method consisting of a few layers of convolutions and MLPs as mentioned by the reviewer on pascalVOC and the result is as below. We noticed that just a simple fusion gives a very bad result but it gets better with a residual connection with the original target image feature (fusion(target image, query image) + target_image). This result demonstrates that the benefits of indirect attention is in fact more than simple feature fusion.
> | Method | Seen Classes | Unseen Classes |
> |---|---|---|
> | Simple Fusion | 26.6 | 30.7 |
> | Simple Fusion + residual | 82.1 | 62.7 |
> | Indirect-Attention | **82.94** | **65.13** |
>
> 3. All results in every table represent averages over 5 runs, each with different randomly selected query images. Though not explicitly said in the experiment section, it has been mentioned in section 5.3. This protocol ensures our findings are stable across query image variations.
>
> 4. Yes, indirect attention is indeed used during pretraining, but with the backbone frozen which is not ready yet for OSOD task. As requested, here are the pretraining-only results (5-run average) on Pascal-VOC:
> | Method | Seen Classes | Unseen Classes |
> |---|---|---|
> | Pretraining only | 11.26 | 14.6 |

---

> > ### Comment · Reviewer_xquQ · 2024-11-26
> > **Respond to the Rebuttal**
> >
> > Dear authors,
> >
> > Thanks for the response, I have read the rebuttal.  The comparison results of "simple fusion + residual" are actually very close to the proposed method, especially in the seen classes. and I believe that the results of both "simple fusion" and  "simple fusion + residual" could be further improved with more proper training of the newly added MLP (since it is totally randomly initialized). Thus, I still have concerns about the design of indirect attention.  I will maintain my initial scores.

---

> ### Author Response · Authors · 2024-11-26
>
> Dear reviewer,
>
> Thanks for your response. The result reported as "simple fusion" + residual is not as basic as it might initially appear. Beyond the incorporation of MLPs specific to the query and target image features, we also introduced additional convolutional blocks for each feature type independently after the fusion stage. All that is in addition to a cross-attention module following it. To clarify further, just the first cross-attention block in the "double cross-attention" setting has been replaced by the MLP + convolutions fusion (simple fusion). So here, we are comparing only one module of indirect-attention with "simple fusion" + a cross-attention module while our indirect-attention module is equivalent only to the cross-attention module part removing the need for "simple fusion" and enhancing the performance at the same time. In the results provided in this table apart from the backbone all other parameters are randomly initialized including the indirect-attention.\
> In addition, the critical performance metric in OSOD is the performance on unseen classes. While the "simple fusion" performs comparably to double cross-attention in seen classes, it significantly underperforms in unseen classes—precisely where our indirect-attention mechanism demonstrates its strength.
> We hope this clarification resolves the concern.

---

### Official Review · Reviewer_GBux · 2024-11-02

**Soundness:** 2
**Presentation:** 2
**Contribution:** 2
**Rating:** 5
**Confidence:** 4

**Summary:**

This work introduces a method for one-shot object detection, a domain that requires training only on base categories without fine-tuning on novel categories. Specifically, the method employs an architecture comprising solely a feature encoder, i.e., a backbone model, and a decoder with indirect attention, which processes queries, keys, and values from distinct sources rather than the same source as in conventional cross-attention mechanisms. Furthermore, a two-stage training approach is utilized, involving pretraining with a hybrid strategy that combines supervised and self-supervised learning, followed by a fine-tuning stage. The incorporation of box-to-pixel relative position bias and contrastive loss enhances performance. Comprehensive experiments on the Pascal VOC and COCO datasets are conducted and  results for both seen (base) and unseen (novel) categories are reported in the work.

**Strengths:**

1. This paper is overall well-written, with just a few typos to address.
2.  Rather than employing a standard architecture with a backbone, encoder, and decoder, the proposed work eliminates the feature encoder for support-query aggregation and introduces a cross-attention mechanism that directly processes the support image, target image, and object queries. This design appears to offer a more generalized solution that facilitates direct application without the need for fine-tuning.

**Weaknesses:**

1. The primary novelty of this paper, the indirect attention mechanism, lacks a clear theoretical foundation. Indirect attention takes Q as the object query, K as the support (query) image feature, and V as the entire image feature, thereby adjusting the feature for each object query based on the global image feature and the similarity between support and object queries. Given that both the support and object queries represent local aspects of an object, it remains unclear how the mechanism determines the channel weights of the image feature, which encompasses multiple objects as well as the background.
2. The results in the ablation studies do not align with those presented in the main table, Table 1. Specifically, the AP0.5 for seen categories in Table 1 is 73.5, whereas the results for seen categories in Tables 3-5 are reported as higher, despite all evaluations being conducted on the Pascal VOC dataset according to line 403. This inconsistency undermines the persuasiveness of the experimental results. In addition, it lacks experimental comparison with recent studies that are published in 2023 and 2024.

**Questions:**

Here are more comments for your concern,

1.  Typos

line 292: 3.5 TRAINING STRATEGY: --> 3.5 TRAINING STRATEGY (no clone here).
Line 246: where the whre --> where the

2.  A curious querstion here: as claimed in line 208-209, "in the decoder, instead of using ...., we proposed indirect attention, that directly exploits ...."  since it is an directly exploitation, why the process is named "indirect"  attention?

3.  What's the query patch in 322, shouldn't the output only contains vectors corresponding to each object query and the background?

4.  In 299, it claimed that this work adopts two-stage training, pretraining and finetuning. As claimed in the related work, one-shot object detection(OSOD) does not allow for a finetuning, so it is unfair to compare with other OSOD methods? Hope the authors can explain more on the finetuning stage.

5.  In line 197, it assumes that "the target image contains at least one instance of the same class as the object in Q", here Q is the query image.  The assumption actually cannot hold in realistic settings, as for detection we don't know the content of the target image, which may or may not include the support(query) class

---

> ### Author Response · Authors · 2024-11-21
>
> 1. We thank the reviewer for the comment. The indirect-attention mechanism operates as a principled extension of transformer attention instead of a heuristic modification. While classical attention presumes alignment between keys and values derived from the same source, our key insight is that such alignment is not obligatory all the times. By decoupling keys and values, and allowing them to originate from distinct sources, indirect-attention enables a more flexible and robust form of feature interaction, particularly advantageous for OSOD.
> Although the precise theoretical underpinnings of this mechanism require further exploration, we attempted to gain insights through the visualizations presented in Section 5.1. Based on our observations, the mechanism appears to operate as follows: Object queries—composed of learnable parameters combined with top-k target image features—serve as a bridge between the query and target images. Specifically, when these queries attend to features in the query image, they activate selectively in regions that correspond to the object of interest. Subsequently, position bias grounds these activated queries within the spatial context of the target image. This interplay likely creates a precise attention flow, where features from the query image guide the selection of relevant regions in the target image via the intermediary object queries.
>
> 2. We thank the reviewer for the comment. We’d like to clarify the apparent discrepancy between Tables 1,2 and 3-5: This reflects experimental design rather than inconsistency. Tables 3-5 use a controlled setting without pretraining and backbone freezing to isolate indirect attention's contribution. This clean-room approach is essential for rigorous ablation studies. Table 6 then systematically reintroduces these components to demonstrate their complementary benefits.
>
> Questions:
> 1. We appreciate the attention to detail on typos, we rectified them accordingly.
> 2. Your question about the "indirect" terminology is insightful. The name reflects how attention between target and query images is mediated through object queries - unlike traditional direct cross-attention between two sequences, we use a third partially learnable sequence (object queries) that orchestrates the interaction, hence "indirect."
> 3. The query patch here is the random crop of the image that is used as a query to extract the position where it belongs in the main image. Yes, that is correct the output just contains either the background or the relevant part of the image. However, since the query patch also belongs to the same image it is mentioned that it does not take part in the loss calculation.
> 4. The term "finetuning" has caused some understandable confusion. To be absolutely clear: both training stages (pretraining and finetuning) train exclusively on seen classes. So our approach maintains strict OSOD constraints throughout.
> 5. The assumption about target images containing query class instances is indeed a general OSOD limitation, which we've acknowledged in our limitations discussion in section 5.2.

---

> > ### Comment · Reviewer_GBux · 2024-11-27
> > **Respond to the Rebuttal**
> >
> > Dear Authors,
> >
> > Thanks to the author's effort in providing a rebuttal. The rebuttal addresses some of my concerns, but I would like to retain my original rating. I still have the following concerns,
> >
> > 1.  Even though the reviews explain the key, value, and query are not necessarily the same. However, it still does not make sense to me how the cross-attention between object queries and support reflects the importance of the image feature. It still needs more theoretical reasoning on this design.
> >
> > 2.  It is hard to decide whether the results are based on a clear setting and solid experiments. Isn't the full model based on the baseline but with BoxRPB and indirect attention? What's the difference between Table 1 and the last row of Table 3 in the experimental setting? In addition, it seems the novel design sacrifices the performance of seen classes and gains on unseen classes. Comparing Table 1 and Table 3, the proposed IA-DETR in Table 1 has dramatically better results on unseen classes, with ~16% improvement, and worse results on seen classes with a ~10% decrease. What causes such a large difference? It should be included in the method and the experimental results, and this information should definitely be considered when evaluating the work.

---

> ### Author Response · Authors · 2024-11-27
>
> Dear reviewer,
>
> Thanks a lot for your response.
> 1. We would be happy to further clarify on how relation between object queries and support features (or query features)  reflects the importance of the image feature.\
> **Considering the first indirect-attention block**: as mentioned previously, in our approach, the object queries—comprising a combination of learnable parameters and the top-k target image features—serve as the query in the indirect-attention mechanism. These queries are multiplied with the query image features (or support image features) to compute the attention scores. Next, a softmax is applied to these attention scores, converting them into probability distributions between 0 and 1 giving higher weights (closer to 1) to object queries which are well-aligned with support image (or query image), and less weights (close to 0) to the object queries irrelevant to the support image features. Why this reflects the importace of target image features? we do not forget that object queries were not only learnable parameters but also **top-k target image features**.\
> Now, you are right that there is nothing significant about target image features up to here apart the top-k target features added to object queries. But, in the next step we add box positional bias(BoxRPB) to these attention scores. The BoxRPB reminds each object query (attention score) where it belongs spatially on the target image features. So the queries (attention scores) are now informed about their relevance to query image features and also about their spatial position in target image features (thanks to the addition of BoxRPB). Multiplying the "informed" object queries with the target image features we predict the boxes and classes.\
> **Subsequent blocks**: going to the next block of indirect-attention, the object queries are already affected by the target image features in last operation of previous block (multiplication with target image features), so again the multiplication with query image features and softmax indeed reflects the importance of target image features at each block. We hope it is clearer now.
>
> 2. Thanks for your detailed look into the experiments section and apologies that it seems a bit confusing at first. The full model is based on  BoxRPB,  indirect attention, early backbone freezing, and contrastive-loss based pretraining. However the main components are the BoxRPB, and indirect-attention which are highly interdependent as well. So the difference between the result in table-1 and the last row of table-3 is the missing early backbone freezing and contrastive-loss based pretraining in table-3. This is because in table-3 our main focus is on ablating BoxRPB, indirect-attention and the interplay between them.\
> The discrepancy between row-3 of table-3 is explained by table-6. In table-6 we add the two additional components (early backbone freezing, and contrastive-loss based pretraining) back and achieve the result in table-1.\
> As you have noticed, there is a huge difference between seen and unseen classes in the ablations part until table-6. This gap is relatively closed mainly by early backbone freezing but as you have mentioned, at the cost of drop of performance on the seen classes. This early backbone freezing prohibits the backbone from overfitting on the seen classes which leads to significant generalization to unseen classes. As per your suggestion we have slightly expanded on this in our new submission.

---

### Note · Authors · 2025-01-22

I have read and agree with the venue's withdrawal policy on behalf of myself and my co-authors.